# Auditory brainstem responses as a biomarker for cognition
Yasmeen Hamza [1,2] ✉, Ye Yang[1], Janie Vu[1], Antoinette Abdelmalek[1], Mobina Malekifar [1], Carol A. Barnes [3] & Fan-Gang Zeng [1] ✉

A non-invasive, accessible and effective biomarker is critical to the diagnosis, monitoring and treatment of age-related cognitive decline. Recent work has suggested a strong association between auditory brainstem responses (ABR) and cognitive function in aging macaques. Here we show in 118 human participants (66 females; age range=18-92 years; hearing loss = -5 to 70 dB HL) that cognition is associated with both age and hearing level, but this triad relationship is mainly driven by the age factor. After adjusting for age, cognition is still significantly associated with both the ABR wave V amplitude (B, 0.110, 95% CI, 0.018– 0.202; $p$ = 0.020) and latency (B, -0.101, 95% CI, -0.186– -0.016; $p$ = 0.021). Importantly, this age-adjusted ABR-cognition association is primarily driven by older individuals and language-dependent cognitive functions. We also perform the area under the curve (AUC) of the receiver-operating-characteristic analysis and find that the ABR wave V amplitude is best for detecting good cognitive performers (AUC = 0.96) whereas the wave V latency is best for detecting poor ones (AUC = 0.86). The present result not only confirms the previous animal work in humans but also shows the clinical potential of using auditory brainstem responses to improve diagnosis and treatment of age-related cognitive decline.

People are not only living longer than ever before but also having to deal with age-related health issues like reduced physical, physiological, and psychological capabilities. Cognition is an important mental ability that allows acquisition, understanding and application of knowledge through perception, sensation, attention, memory, language and other high-level processing[1]. Cognition is highly correlated with age but also affected by other factors like genetics, sensory loss, trauma, education, exercise and other lifestyle choices[2,3]. A prime example of importance in cognition is Alzheimer's Disease (AD) and related dementia, which impair cognitive ability and currently affects 55 million people worldwide with an annual economic burden of US$1.3 T[4]. An accurate, reliable, and accessible biomarker for cognition would help not only monitor normal aging and lifestyle choices but also detect and treat AD and related dementia[5-7].

At present, biomarkers that can predict changes in cognition during normative aging are scarce, with most biomarkers being specific to the pathophysiological processes like blood or cerebrospinal fluid concentrations of Amyloid β1–42, total tau, and phosphorylated tau[8-11]. Some of these biomarkers include magnetic resonance imaging (MRI) that can assess hippocampus volume, and positron emission tomography (PET) that can assess glucose hypometabolism, amyloid or tau buildups in temporoparietal regions[12]. These biomarkers have played a crucial role in defining and understanding both normative cognitive decline and AD, but they are either invasive or expensive or both, limiting their utility in frequent testing and longitudinal monitoring[8].

To overcome these limitations, monitoring the electrophysiological activity of the nervous system has been proposed as an additional sensitive biomarker. Electrophysiological measures are noninvasive and cost-effective, and importantly have high temporal resolution, allowing detection of subtle changes in neurotransmission that may indicate the onset or progression of a neurodegenerative process[13,14]. One such change is tau deposition in the nervous system that can cause synaptic loss, axonal retraction, or eventually cell death[15]. Because tau deposition occurs in subcortical structures that send input to the cortex, monitoring subcortical electrophysiological activities may hold promise for early detection of age-related cognitive decline[16-19]. Indeed, a recent study by Gray et al.[20] found in aging rhesus macaques that slower temporal processing in the auditory brainstem is associated with poorer cognitive function. Two questions have naturally arisen from this discovery. First, why is an auditory measure associated with cognitive function? Second, can this association be extended to humans?

[1]Center for Hearing Research, Otolaryngology-Head and Neck Surgery, University of California Irvine, Irvine, CA, USA. [2]Institute of Sound and Vibration Research, School of Engineering, University of Southampton, Southampton, UK. [3]Psychology, Neurology and Neuroscience, and Evelyn F. McKnight Brain Institute, University of Arizona, Tuscan, AZ, USA. ✉e-mail: y.hamza@soton.ac.uk; fzeng@uci.edu

There is a well-established association between hearing loss and cognitive decline[21–23]. Not only is hearing loss the number one modifiable risk factor for dementia in midlife[24], but also hearing aid use may decrease dementia risk in susceptible older individuals[25,26]. This positive result with hearing aid use is important because clinical trials with other modifiable risk factors such as exercise or mindfulness training have produced mixed findings[27,28].

Although the underlying neurophysiological mechanisms relating hearing loss to cognitive decline are unclear, their strong relationship suggests the potential beneficial use of auditory brainstem responses (ABR) in monitoring human cognitive function. Interestingly, the ABR is likely the most used objective measure in audiology and otology from newborn universal hearing screening to differential diagnosis of lesion sites[29–32]. The ABR involves a simple electrode montage of just 2–4 surface electrodes and passive responses that can be conducted even during sleep[33]. The ABR waveform contains up to five vertex-positive peaks that occur in the first 10 ms from the onset of a transient sound like a click. The neural generators of the ABR are also well described, with wave I being from the auditory nerve, and wave V from the lateral lemniscus and inferior colliculus[34]. The peak latency reflects synaptic delay and spike velocity, whereas the amplitude reflects the total number of excited neurons and the spike synchrony across these neurons[35]. Generally, relatively high-level and high-rate stimuli are preferred to reveal subtle changes in neurotransmission from cochlear synaptopathy to central lesions[36,37].

To extend the Gray et al. finding in nonhuman primates to humans, we collected the ABR to a 60-dB SL, 51-Hz click train and cognitive performance in ten different measures in 118 adults. The wide range of age and various degrees of hearing level in these adults allowed us to delineate the age and hearing factors (Fig. 1A) from the ABR-cognition relationship (Fig. 1B, C). We also hypothesized that the ABR wave V, not wave I, would reflect subcortical neurological changes that are associated with corresponding cortical and cognitive changes.

## Results

Table 1 shows the participant characteristics and test scores in the full sample and three age groups (young, middle-aged and elderly). On average, the hearing threshold was significantly increased from 1.39 dB HL PTA in the young group to 11.30 dB in the middle-aged group and 22.44 dB in the elderly group ($F_{2,115} = 25.791$, $p < 0.001$). This age-dependent hearing threshold elevation is partially consistent with suprathreshold speech recognition measures. While all age groups achieved nearly perfect (>98%) speech recognition in quiet ($F_{2,113} = 0.487$, $p = 0.616$), the younger the age, the better performance in speech recognition in either the steady-state noise ($F_{2,113} = 4.997$, $p = 0.008$) or a competing talker background ($F_{2,113} = 15.813$, $p < 0.001$). The remaining section focuses on cognitive performance, with domain-specific cognitive performance being displayed in Table 2.

### The triad relationship of age, hearing and cognition

The univariate analysis showed, not surprisingly, that cognitive performance is significantly associated with age (Fig. 2A: B, −0.021; 95% CI, −0.025 to −0.017; $p < 0.0001$) and hearing level (Fig. 2B: B, −0.021; 95% CI, −0.027 to −0.014; $p < 0.0001$). On average, each 10-year increase in age decreases the cognitive composite $z$-score by 0.21, while each 10-dB increase in PTA decreases that by 0.27. Note that this significant association revealed by the univariate analysis is uncorrected for covariates and is included to represent the raw data.

The multivariate analysis showed that age is still significantly associated with cognition after adjusting for hearing level (Fig. 2C: B, −0.020; 95% CI, −0.025 to −0.015; $p < 0.0001$), but, surprisingly, the hearing level is not associated with cognition after adjusting for age (Fig. 2D: B, −0.003; 95% CI, −0.010 to −0.004; $p = 0.385$). This multivariate result suggests that the triad relationship is mainly driven by the age-cognition association.

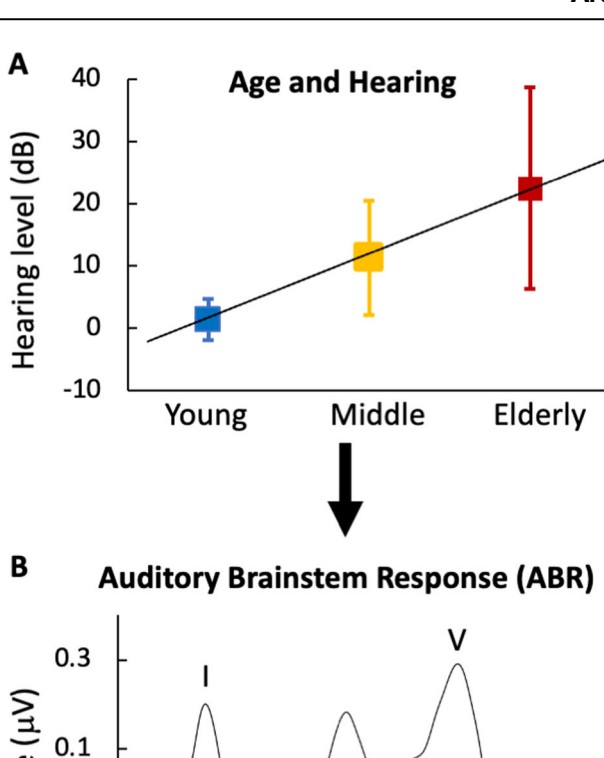

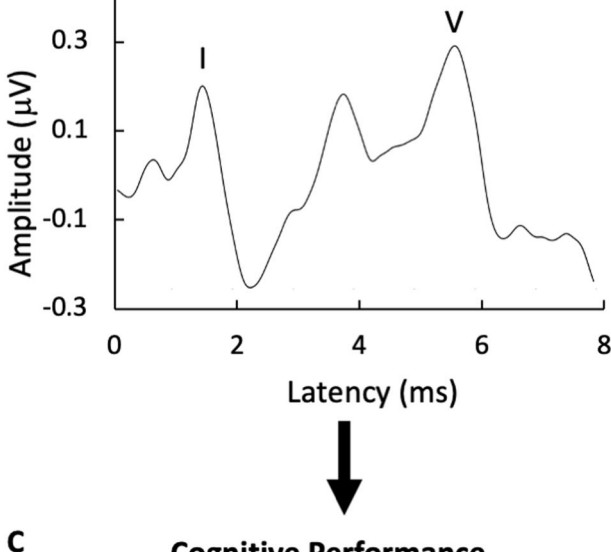

**Fig. 1 | Schematic approach to delineating the assocation between the auditory brainstem response or ABR and cognitive performance.** Adjusting age and hearing (**A**) to reveal the association between the ABR (**B**) and cognitive performance (**C**). Hearing Level refers to pure tone average across the frequencies 0.5, 1,2, and 4 kHz in the better ear. The square represent the mean and the error bar the 95% confidence interval (blue = Young; yellow = Middle; red = Elderly). The ABR is recorded with a high stimulation rate of 51 Hz. The ABR wave I and V latency or amplitude is used to predict cognitive performance in either an overall $z$-score or 10 different measures including four language-dependend ones from CERAD (Consortium to Establish a Registry for Alzheimer's Disease) and six non-language-dependent measures.

## Table 1 | The participant characteristics and outcome measures

| Variable | Full-sample | Young $n$ = 26 | Middle-Aged $n$ = 26 | Elderly $n$ = 66 |
|---|---|---|---|---|
| **Age, mean (SD), range, year** | **55.68 (20.97), 18–92** | **23.00 (3.51), 18–30** | **49.69 (7.70), 31–59** | **70.92 (9.34), 60–92** |
| Gender | | | | |
| Females, No. (%) | 66 (55.93) | 14 (53.85) | 15 (57.69) | 37 (56.06) |
| Males, No. (%) | 52 (44.07) | 12 (46.15) | 11 (42.31) | 29 (43.94) |
| PTA, mean (SD), range, dB HL | 15.35 (15.60), −3.75–70 | 1.39 (3.28), −3.75–7.5 | 11.30 (9.12), −1.25–30 | 22.44 (16.31), −2.5–70 |
| HINT-Quiet word, mean (SD), range, % correct | 99.46 (1.32), 93.33–100 | 99.41 (1.44), 93.40–100 | 99.60 (1.08), 95.00–100 | 99.42 (1.37), 93.33–100 |
| HINT-Quiet sentence, mean (SD), range, % correct | 98.21 (4.59), 70.00–100 | 98.21 (4.59), 70.00–100 | 98.94 (2.18), 93.55–100 | 98.14 (4.55), 75.00–100 |
| HINT SSN, mean (SD), range, dB SNR | 1.38 (3.25), −9.33–14.33 | -0.04 (2.91), −9.33–5.00 | 0.84 (2.82), −4.33–7.00 | 2.16 (3.34), −3.00–14.33 |
| HINT-Male Speaker, mean (SD), range, dB SNR | 0.39 (6.29), −13.00–16.60 | -4.41 (4.98), −12.33–4.33 | -0.78 (6.23), −12.33–16.60 | 2.75 (5.60), −13.00–16.33 |
| **Composite cognition, mean (SD), range, *z*-score** | **0.00 (0.64), −1.99–1.12** | **0.60 (0.33), −0.14–1.12** | **0.14 (0.45), −1.01–1.07** | **-0.29 (0.61), −1.99–0.88** |
| ABR Wave I Latency, mean (SD), range, ms | 1.80 (0.26), 1.23–2.94 | 1.79 (0.18), 1.46–2.27 | 1.79 (0.25), 1.44–2.31 | 1.80 (0.29), 1.23–2.94 |
| ABR Wave I Amplitude, mean (SD), range, μA | 0.10 (0.09), 0.01–0.64 | 0.11 (0.07), 0.01–0.28 | 0.12 (0.14), 0.01–0.64 | 0.09 (0.70), 0.01–0.32 |
| ABR Wave V Latency, mean (SD), range, ms | 6.05 (0.30), 5.31–6.87 | 5.95 (0.21), 5.48–6.39 | 6.01 (0.36), 5.43–6.64 | 6.11 (0.30), 5.31–6.87 |
| ABR Wave V Amplitude, mean (SD), range, μA | 0.23 (0.11), 0.00–0.55 | 0.30 (0.11), 0.08–0.50 | 0.25 (0.10), 0.07–0.41 | 0.19 (0.09), 0.00–0.55 |
| ABR I-V Latency Difference, mean (SD), range, ms | 4.25 (0.27), 3.69–5.04 | 4.15 (0.18), 3.81–4.43 | 4.22 (0.27), 3.69–5.04 | 4.29 (0.29), 3.73–5.00 |
| ABR V/I Amplitude Ratio, mean (SD), range, μA | 4.54 (5.78), 0.0–36.68 | 4.94 (7.18), 0.76–36.68 | 4.93 (4.93), 0.59–23.8 | 4.22 (5.54), 0.00–29.8 |

## Table 2 | The participants' cognitive specific domain scores

| Cognitive Test mean (SD), range | Full-sample | Young $n$ = 26 | Middle-Aged $n$ = 26 | Elderly $n$ = 66 |
|---|---|---|---|---|
| Word Learning | 22.10 (3.59), 12–29 | 24.08 (2.54), 19–29 | 22.92 (3.31), 15–27 | 21.00 (3.66), 12–28 |
| Delayed recall | 7.08 (2.01), 1–10 | 8.42 (1.33), 6–10 | 7.31 (1.89), 3–10 | 6.47 (2.02), 1–10 |
| Word Recognition | 19.52 (1.31), 10–20 | 19.92 (0.39), 18–20 | 19.81 (0.40), 19–20 | 19.24 (1.67), 10–20 |
| Animal Fluency | 22.81 (5.65), 9–36 | 25.73 (5.58), 11–34 | 23.31 (4.57), 14–32 | 21.47 (5.66), 9–36 |
| TMT A (Executive), seconds | 26.77 (8.84), 12–57 | 20.48 (4.70), 12–32 | 25.15 (7.29), 15–39 | 29.89 (9.22), 14–57 |
| TMT B (Executive), seconds | 66.65 (27.33), 30–218 | 50.35 (13.72), 30–84 | 61.62 (21.42), 36–134 | 75.05 (30.13), 37–218 |
| 4 MT accuracy (spatial) | 10.10 (2.87), 2–15 | 11.85 (2.11), 5–15 | 10.50 (2.87), 2–14 | 9.26 (2.82), 3–14 |
| 4 MT RT (spatial), mseconds | 8150.77 (2221.63), 3968–13810 | 7421.54 (2033.72), 4007–11627 | 7538.85 (1736.56), 3968–10305 | 8679.11 (2342.56), 4474–13810 |
| Visual Discrimination | 15.97 (1.82), 11–18 | 17.27 (0.96), 14–18 | 15.92 (2.04), 11–18 | 15.48 (1.76), 11–18 |
| SDMT (Attention/working memory) | 52.42 (11.63), 21–79 | 64.54 (9.24), 40–79 | 52.73 (8.94), 35–72 | 47.53 (9.84), 21–66 |

### ABR wave V is associated with cognition even after age adjustment

The univariate analysis showed that cognitive performance is significantly associated with wave V latency (Fig. 3A: B, −0.212, 95% CI, −0.322 to −0.101; $p < 0.0001$) and wave V amplitude (Fig. 3B: B, 0.288, 95% CI, 0.183–0.392; $p < 0.0001$). Even after age adjustment, this significant association with cognition is still maintained for both wave V parameters (Latency in Fig. 3C: B, −0.101, 95% CI, −0.186 to −0.016; $p = 0.021$; Amplitude in Fig. 3D: B, 0.110, 95% CI, 0.018–0.202; $p = 0.020$). Note though that the age adjustment decreases the effect size (B value) by 52% for the wave V latency and 62% for the wave V amplitude.

Four additional observations are worth noting. First, wave I (amplitude or latency) is not associated with cognition (Supplementary Table 1) or hearing level (Supplementary Table 2). Second, hearing level is not associated with ABR wave V latency, and like the cognition result, the association of hearing level with the ABR wave V amplitude can be fully accounted for by age (Supplementary Table 2). Third, this age-adjusted ABR-cognition association is mostly driven by the language-dependent cognitive tasks (e.g., wave V latency is associated with the CERAD word learning, delayed recall, and animal fluency tests, see Supplementary Table 3; wave V amplitude with the word learning and animal fluency tests plus a task on attention and working memory, see Supplementary Table 4). Finally, because the two combined measures (I–V latency difference and V/I amplitude ratio) produced inconsistent associations (Supplementary Tables 1 and 2), they were not used as a potential biomarker for cognition.

### The elderly participants drive the overall cognition associations

We repeated the above analysis in all three age groups. Indeed, the univariate analysis revealed only significant association between cognitive performance and age in the older age groups (middle-aged: B, −0.026; 95% CI, −0.048 to −0.04; $p = 0.021$; elderly: B, −0.037; 95% CI, −0.050 to −0.24; $p < 0.0001$) as well as that between cognitive performance and hearing level (middle-aged: B, −0.022; 95% CI, −0.040 to −0.004; $p = 0.021$; elderly: B, −0.019; 95% CI, −0.019 to −0.001; $p = 0.033$). No such significant associations were found in the young group ($p > 0.05$). Consistent with the overall result, age adjustment abolished the significant association of hearing level with cognition in both middle-aged and elderly groups ($p > 0.05$).

We also categorized the hearing level variable into either normal hearing (PTA < 20 dB HL) or hearing loss (PTA ≥ 20 dB HL) to perform a similar sensitivity analysis. None in the young group had hearing loss. This binary categorization of hearing level showed significant association with cognition in only the elderly group (B, −0.304 95% CI, −0.595 to −0.012;

**Fig. 2 | Triad relationship between age, hearing and cognition.** Univariate analysis on correlating cognitive performance with age (**A**), and hearing level (**B**). Multivariate analysis of the same correlation with age after adjusting for hearing level (**C**), and with hearing level after adjusting for age (**D**). Individual data are separated by age: blue circles = young, yellow circles = middle-aged, red circles = elderly. The dashed line represents linear regression with significant regression ($p < 0.05$) having an arrow at the end and *.

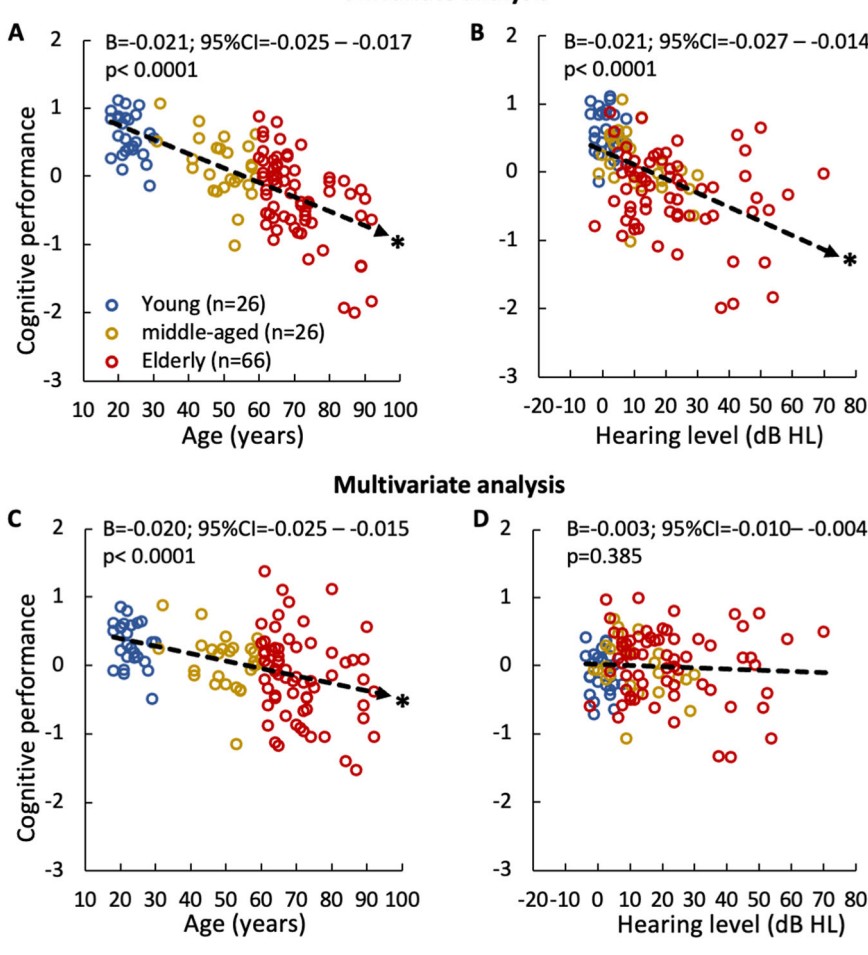

$p = 0.041$), and age adjustment within the elderly group also abolished this significant association ($p = 0.901$).

Finally, sensitivity analyses showed that cognitive performance was significantly associated with wave V latency (B, −0.192; 95% CI, −0.336 to −0.048; $p = 0.010$) and wave V amplitude (B, 0.175; 95% CI, 0.006–0.344; $p = 0.042$) only in the elder group. However, age adjustment reduced the significant association to a trend in the elderly group (latency: $p = 0.067$; amplitude: $p = 0.056$). This reduced significance level is consistent with the observed reduction in effect size in the overall result.

### ABR can predict cognitive performance, especially in low and high performers

Figure 4A shows systematically how well the ABR can detect cognitive performance above or below a fixed criterion from the cognitive z-score 1 to 99 percentile. The area under the curve (AUC) is significantly above the chance performance (mean = 0.5, std = 0.02; gray dashed line) for both wave V parameters. For wave V latency (orange line), the mean is 0.67 (std = 0.06, range = 0.59–0.86, $p < 0.0001$); for wave V amplitude (purple line), the mean AUC is 0.72 (std = 0.06, range = 0.52–0.96, $p < 0.0001$). Note that the highest AUC values occur at either the low or high end of cognitive performance. The highest AUC (0.86) is for wave V latency to detect cognitive performance below the 3rd percentile (open orange circle), whereas that (0.96) is for wave V amplitude to detect performance above the 98th percentile (open purple circle). At these highest AUC values, using the >6.42-ms criterion, the ABR wave V latency can detect low cognitive performers below 3-percentile with 75% sensitivity and 90% specificity (open orange circle in Supplementary Fig. 1A); using the >0.39-μV criterion, the wave V amplitude can detect high cognitive performers above 98-percentile with 94% sensitivity and 100% specificity (open purple circle in Supplementary Fig. 1B).

Figure 4B shows the same results as Fig. 4A, except that the cognitive performance is adjusted for age. The age-adjusted AUC for both wave V parameters was above the chance performance but slightly smaller than the unadjusted values (latency: mean ± std = 0.63 ± 0.05 vs. 0.67 ± 0.06, $p < 0.001$; amplitude: 0.70 ± 0.07 vs. 0.72 ± 0.06, $p = 0.03$). Importantly, the same trends are preserved for the two parameters' predictability of the cognitive performance even after the age adjustment: The highest AUC (0.71) is for wave V latency to detect cognitive performance below 3-percentile (orange circle), whereas that (0.82) is for wave V amplitude to detect performance above 93-percentile (purple circle). The highest AUC values were obtained for wave V latency >6.24-ms (Supplementary Fig. 1C) and wave V amplitude >0.25-μV (Supplementary Fig. 1D).

Figure 5 shows the 2-fold cross validation result, while 5- and 10-fold results are included as Supplementary Figs. 2 and 3. Overall, the cross validation produced a remarkably similar pattern of results to the classical ROC analysis: ABR wave V latency was better for detecting poor cognitive performers while its amplitude better for good performers. On average, the 2-fold cross-validation AUC was similar to the classical one: raw wave V latency (0.66 ± 0.07 vs. 0.67 ± 0.06, $p = 0.04$); raw amplitude (0.71 ± 0.09 vs. 0.72 ± 0.06, $p = 0.68$); age-adjusted latency (0.60 ± 0.09 vs. 0.63 ± 0.05, $p = 0.02$); age-adjusted amplitude (0.70 ± 0.10 vs. 0.70 ± 0.07, $p = 0.87$).

### Discussion

Here we assessed the triad relationship between age, hearing, and cognition in a large human population ($n = 118$). The present study found that age can fully explain the hearing-dependent association with cognition, but not vice versa (Fig. 2). The present study also found that a synchronized neural response in the auditory brainstem (wave V) is associated with cognitive performance, especially in language-dependent measures. Importantly, this

**Fig. 3 | Significant correlation between ABR and cognitive performance.** Univariate analysis on correlating cognitive performance with wave V latency (**A**) and amplitude (**B**). The age-adjusted correlation with wave V latency (**C**) and amplitude (**D**). All units are in *z*-scores. All symbols and lines are the same as Fig. 2.

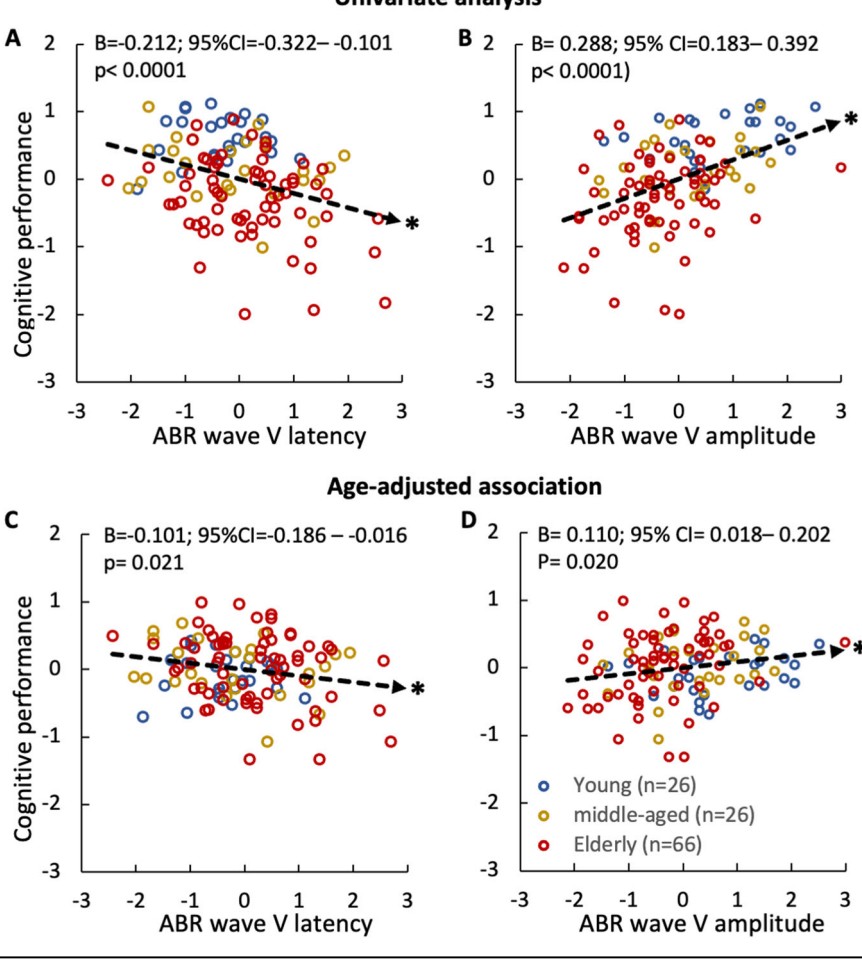

ABR-cognition association persists even after age adjustment (Fig. 3). Systematic receiver-operating-characteristic curve analysis showed that both ABR wave V latency and amplitude can serve as an age-independent biomarker for cognition, with the latency best detecting low performers while the amplitude detecting high performers (Fig. 4). Cross validation further verified this ABR-cognition association (Fig. 5). The overall result suggests that the auditory brainstem response is correlated with two components of cognition, with one being age-dependent and the other being age-independent.

### ABR-cognition association is independent of age and hearing loss

Traditionally, aging involves two different processes: sensory degeneration in the cochlea[38] and neural atrophy in the brain[39]. The cochlear degeneration may produce hearing loss detected by audiograms, but central atrophy may not[40,41]. Therefore, even though age and hearing loss covary, age is associated with cognition after adjusting for hearing loss but not vice versa. Moreover, the present result found that the ABR-cognition association still exists even after adjusting for both a wide range of age and various degrees of hearing loss. This finding is important because it suggests that not only does the ABR-cognition association go beyond age and hearing loss, but also the ABR may be able to serve as a biomarker for cognition in the general population.

### ABR-cognition is specific to cognitive domains

The present study also found that the ABR wave V parameters are associated with language and memory-related functions in the medial temporal lobe, but not with visual discrimination and executive functions in the occipital and frontal lobes (Supplementary Tables 3, 4). On the one hand, this domain-specific association with wave V contrasts with the lack of

significant association between the hearing level and age-adjusted cognitive performance, overall or any specific domain (Supplementary Table 5). On the other hand, this medial temporal lobe-specific association is strikingly similar to the pattern of results observed in aging macaques[20]. These results suggest that individual differences in neural structure and connection in the temporal lobe are responsible for the individual variability in cognitive performance in the normal population.

However, it is not clear why the ABR wave V parameters can predict individual cognitive performance. On surface, the neural structures underlying the ABR wave V and language and memory-related cognitive functions are entirely different, with the former being the lateral lemniscus and inferior colliculus[34] and the latter being the temporal lobe and hippocampus[42,43]. There are likely common neural mechanisms from the brainstem to cortex that are responsible for this ABR-cognition association. We propose two possible such mechanisms based on the present result showing that the wave V latency is a better predictor of poor cognitive function, but the wave V amplitude predicts high. One mechanism is related to synaptic delay and demyelination that produce longer wave V latency and poorer cognitive performance, whereas the other mechanism is related to total neuronal health and synchrony that produce higher wave V amplitude and better cognitive performance[35]. Future investigation is needed to clarify this ABR-cognition relationship and its underlying neural mechanisms.

Because nonhuman primates do not contract AD, the changes observed in these animals are a result of normative aging. In human populations who are susceptible to AD, these ABR changes may be even more apparent, contributing to a decline in learning, memory, and eventually overall cognitive capacity[44,45]. This is at least consistent with the observation of protection of cognition observed in the Lin et al. study[26], in which, even though the two study populations began in the normal cognitive

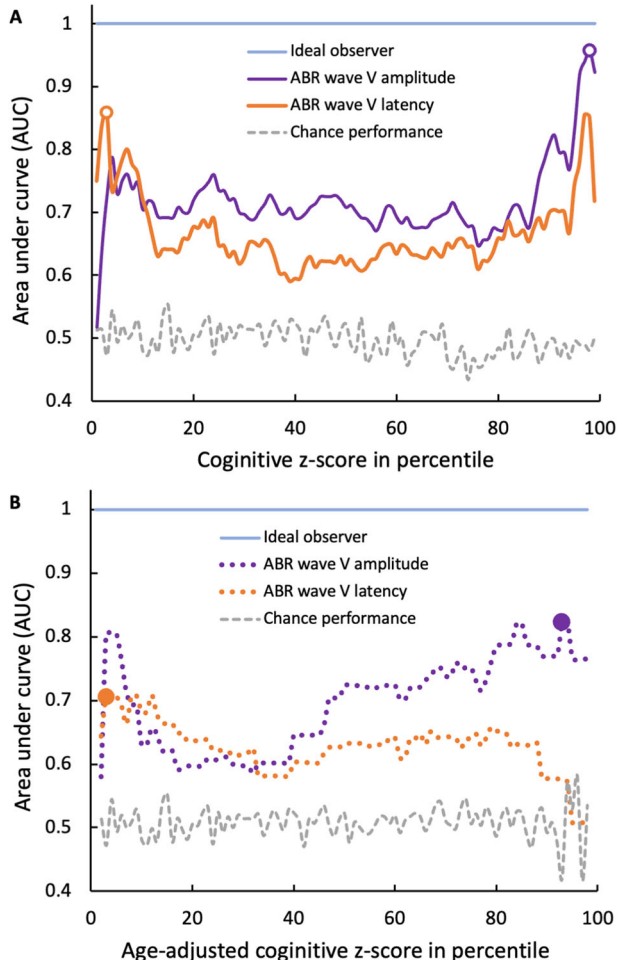

**Fig. 4 | Area-under-curve (AUC) analysis. A** The AUC as a function of cognitive performance using either the ABR wave V amplitude (purple line) or latency (orange line) as a biomarker. The highest AUC value is shown as the open purple circle for the amplitude and the orange open circle for the latency. Ideal performance (AUC = 1) is shown as the horizontal blue line, while chance performance (mean AUC = 0.5) from randomly generated ABR parameters as the grey dashed line. **B** The same representation as (**A**), except that the cognitive performance was adjusted for age (*x*-axis).

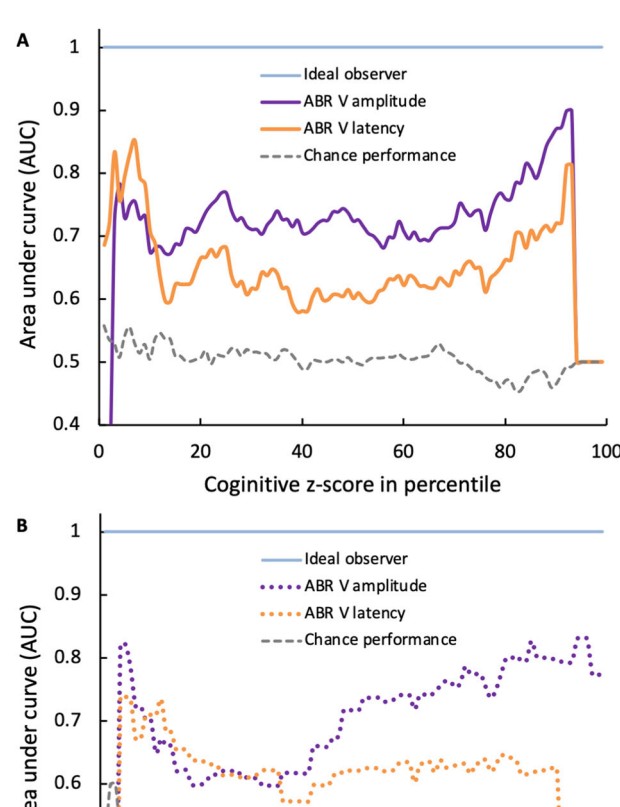

**Fig. 5 | Area-under-curve (AUC) analysis using 2-fold cross validation.** Same as Fig. 4 except that the AUC vs cognitive performance function was generated using 2-fold cross validation. **A** Age-unadjusted cognitive performance (*x*-axis). **B** Age-adjusted cognitive performance (*x*-axis).

range, the population that had more risk factors for AD were the individuals who benefited the most over the three-year intervention trial period. ADRD is associated with the accumulation of tau protein in the medial temporal lobe[46–50], which plays an important role in the decline of episodic memory function[51]. Importantly, the brainstem shows tau depositions well before cortical involvement[16–19,52,53] and is associated with the earliest clinical phenotypes of AD[54,55]. It would be interesting for future studies to examine longitudinally whether hearing aid use would slow down ABR deterioration, corresponding to its positive impact on cognitive decline.

### ABR as a cognitive biomarker: technical considerations

Two carefully selected stimulus parameters likely contributed to the present significant association between ABR and cognition: presenting a relatively high-rate (51 Hz) click train at a fixed 60 dB sensation level. The use of high-rate, high-level click trains likely stressed the neuronal system to reveal subtle differences in temporal processing that affect cognitive performance. In the aging macaque study[20], Gray et al. used a relative wave V latency difference between the 20 and 50 Hz click trains to reveal the significant association between auditory temporal processing and cognitive performance. The present study found that the absolute 51-Hz wave V latency is sufficient to reveal the positive association, saving half of the data collection

time. In addition, Gray et al. used a fixed 60-dB peak SPL, whereas the present study varied the stimulus level to achieve a fixed 60-dB sensation level, or 60 above the behavioral hearing threshold. As a result, the present study found no association between hearing level and wave V parameters. It is also possible that the lack of association between hearing level and wave V amplitude is due to age-related changes that are not reflected by thresholds, e.g., cochlear synaptopathy[56]. Importantly, the current study found that wave V amplitude is associated with cognitive performance, especially in differentiating the high performers. In practice, both latency and amplitude measures need to be explored to screen and monitor ADRD.

### ABR as a cognitive biomarker: clinical utilities

Early detection of age-related cognitive decline is critical to its prevention and management. Clinically, ABR is not only a non-invasive, objective, and cost-effective measure but also readily available, user-friendly, and easily implemented[57]. For example, hearables and wearables may have an online ABR capability that can routinely screen individuals with high-risk factors. Unlike behavioral tasks, the ABR is an objective measure that can be used repeatedly without worries about learning, practice, or other side effects, making it an ideal biomarker for longitudinal studies. Because the ABR can be obtained without attention, during sleep, or under sedation, it is especially useful for monitoring patients whose behavioral tasks are difficult or impossible to measure.

One limitation of the present study is that it does not directly address mild cognitive impairment or ADRD. Although the present study did not

include any clinically diagnosed patients with mild cognitive impairment or ADRD, we found two participants to have mild cognitive impairment based on previously published normative cognitive data[58]. Both participants were in the elderly group (84 and 92 years) and had wave latency (6.47 and 6.87 ms, respectively) longer than the presently proposed 6.42-ms criterion for identifying poor cognitive performers. Another limitation is that the present study did not consider other potentially important variables like sex, head size, education and socioeconomic status. A much larger sample size than the current one would be needed to delineate the contributions of these variables to the ABR-cognition relationship.

In a relatively small sample consisting of 15 elderly normal controls and eight age- and hearing-matched participants with mild cognitive impairment, Bidelman and colleagues found that the brainstem component of the neural frequency-following responses to speech is a more robust predictor of individuals' cognitive impairment than the cortical component[59]. This previous result is highly consistent with the present result, but they both need to be extended to a large sample of actual patients to determine the utility of the ABR test in the detection and monitoring of ADRD. Like the universal newborn hearing screening, the ABR stimulus parameters and electrode montages need to be optimized and standardized for improved diagnostic accuracy and routine monitoring efficiency. Moreover, combining the ABR with imaging studies could help understand the causal relationship between the structural and functional changes. Finally, comparing the ABR between ADRD and other specific temporal lobe deficits as aphasia would reveal global vs. local differences, improving the diagnosis of different neurological disorders.

## Methods

### Participants

One hundred thirty adults participated in this study. Twelve participants were excluded because of incomplete or recently performed cognitive testing ($n = 3$), inability to obtain ABR ($n = 8$, due to claustrophobia, bleeding tendency, wax, or time constraints), and conductive hearing loss ($n = 1$). The resulting 118 participants included 66 females and had a mean age of 56 years (SD = 21). They were classified into three age categories: young (18–30 y; $n = 26$), middle-aged (31–59 y; $n = 26$), and elderly (60–92 y; $n = 66$). The elderly participants were recruited without clinically diagnosed mild cognitive decline or ADRD. The three age groups resembled the age grouping in previous normative data[58], allowing visualization of interactions between the age and other factors. Written informed consent was obtained from all participants at the beginning of the study. The experimental protocol was approved by the University of California Irvine Institutional Review Board. All ethical regulations relevant to human research participants were followed.

### Procedures

Hearing Level. An otoscopic examination was performed to rule out external and middle ear abnormalities in all participants. Pure-tone audiometry was conducted using a Grason-Stadler GSI 61 audiometer and TDH headphones in a double-walled soundproof room. The participant's threshold was determined for the octave frequencies between 0.125 and 8 kHz. The pure-tone average (PTA) over 0.5, 1, 2, and 4 kHz was used to determine the better ear, which would be tested in all subsequent auditory tests. The otoscopic examination and audiological tests took 10–20 min to complete.

Speech Recognition. The participants listened to a list of 10 meaningful sentences consisting of 4–5 keywords (e.g., A boy fell from the window; They went on vacation) in quiet, steady-state noise or a competing voice[60,61]. Speech recognition in quiet was measured as either the overall keywords correctly identified, or the overall sentences correctly identified, which required all keywords in a sentence to be correctly recognized. Speech recognition in noise or the competing voice was measured as the signal-to-noise ratio, at which 50% of the sentences were correctly recognized.

Auditory Brainstem Responses. Auditory brainstem responses to clicks were collected from the better ear using the Bio-Logic Navigator Pro (Natus Medical Inc., Middleton, WI, USA). The click was 100 μs in duration and presented in alternating polarity at a high rate of 51.33 Hz via an ER-3a insert phone (Etymotic Research, Inc., Elk Grove Village, IL). To minimize the effect of hearing loss, the perceptual detection threshold was determined for the click train, and the click level was individually set at 60 dB above the detection threshold (corresponding to 65–90 dB nHL). This relatively high click level was at or above the 60-dB knee point where the ABR wave V parameters became saturated with a further increase in stimulus level[62]. During the ABR recording, the participants were seated in a reclining chair in a double-walled soundproof room and instructed to relax or sleep if possible. Recordings were obtained using insert phones with gold tiptrodes (Etymotic Research, Inc., Elk Grove Village, IL), which served as the inverting electrode at the test ear and ground electrode at the contralateral ear, with the non-inverting electrode being placed on the high forehead (Fz). Electrode impedances were maintained at <3 kΩ. Epoch window was 10.66 ms. Individual recordings exceeding 23.8 μV were rejected from the average, and at least 2000 artifact-free sweep responses were obtained. The ABR recording took about 40 min to complete.

ABR waveforms were bandpass filtered (100–3000 Hz), then averaged for final analysis. Initially, a computer program estimated wave I as the peak between 1.3–2.3 ms and wave V between 5.1–6.4 ms. After that, wave peaks and troughs were manually confirmed and adjusted, if necessary, e.g., a peak outside the range, two peaks in the same time interval, or the lack of a clear peak, by the first author and confirmed and agreed upon by the second author. The wave amplitude was measured from the peak to the following trough. The latency was the duration between the onset of the click and the peak time. The I–V inter-peak latency and V/I amplitude ratio were also calculated as a relative measure to minimize individual differences in sex, age, and head geometry[63,64].

Cognitive Testing. The participants completed eight cognitive tests that yielded 10 outcome measures in the following order. First, to test immediate learning memory, the participant was presented in a written form with 10 unrelated common words from the Consortium to Establish a Registry for Alzheimer's Disease (CERAD)-word learning test[65], recalling as many words as possible, with the correctly recalled words being the outcome measure.

Second, to test executive function, the participant connected a series of numbered (Trail A) and numbered alternating with lettered (Trail B) circles in ascending order as quickly as possible[66]. The completion time was the outcome measure in both trail-making tests.

Third, to test working memory, processing speed and attention, the participant performed the Symbol Digit Modality Test or SDMT[67], in which she or he used a coded key to match nine abstract symbols paired with numerical digits within 90 s. The number of correctly matched pairs was the outcome measure.

Fourth, immediately after the Trail and SDMT tests that typically lasted for five minutes, the participant was asked to recall as many words as possible from the original 10-word list.

Fifth, the participant was also asked to recognize these 10 words mixed with 10 different words using a 20-item forced-choice procedure (i.e., answering either 'Yes' or 'No' to the question: of whether a word occurred in the original list). The number of correctly recognized words was the outcome measure.

Sixth, to also test executive function but in a verbal domain, the participant named as many animals as possible in one minute, i.e., CERAD-Animal Fluency test[65].

Seventh, the participant performed the visual discrimination test from the Neuropsychological Assessment Battery[68], in which she or he matched a target visual design from an array of four similar designs presented beneath the target.

Finally, to test spatial short-term memory, the participant performed the 4-Mountains Test[69], in which she or he matched a target landscape from 4 mountains with a shifted viewpoint on an iPad. Both accuracy and reaction time of the correct responses were used as the outcome measures.

The participants performed the cognitive tests in a quiet conference room and took breaks as needed. The total time to complete these tests was 40–60 min, depending on the individual.

## Statistics and reproducibility

The 10 cognitive outcome measures from eight tests were z-score-normalized for each measure and adjusted for direction so that higher values correspond to better performance. A cognitive composite z-score was obtained as the average over the 10 measures. The hearing level was the PTA threshold in dB HL at 0.5, 1, 2, and 4 kHz in the better ear. The higher the PTA value, the worse the hearing. Wave I and V latency (ms) and amplitude (µV) served as the ABR outcome measures. The wave V/I amplitude ratio was log-transformed to conform to normal distribution, which was confirmed for other measures. Linear regression was performed to assess the association between auditory measures (independent variables) and cognitive performance (dependent variable). All parameters were z-scored. Additionally, analyses were repeated with the individual cognitive test scores to identify task-specific relationships. In all models, the assumptions of linearity, normality of errors, multi-collinearity, and homoscedasticity were met. In multivariate analysis, age was adjusted to test if age could explain the covariation between auditory measures and cognitive outcomes. Except for descriptive statistics reported in Table 1, age was analyzed as a continuous variable in all reports, including the sensitivity analysis that focused on the elderly group. Age effects were also examined using age-groups in General Linear Models with post-hoc Bonferroni correction for multiple comparisons. In all cases, the significance criterion was adjusted to be $p < 0.05$.

Because of the lack of data regarding the association of ABR parameters with cognition in humans, we could not run traditional power analysis to determine a sample size. It is generally accepted that there should be at least 10 observations per independent variable in a regression analysis. Since we used no more than two independent variables in any given model, both the smallest sub-sample size ($n = 26$ in the young and middle-aged groups) and the full-sample size ($n = 118$) should be appropriate and sufficient.

To characterize the association between ABR measures and cognitive performance, the area under the curve (AUC) was calculated from receiver operating characteristic (ROC) curves. The ROC curve was empirically constructed as a function of the percentile in cognitive performance from 1% to 99%, in 1% steps. For each percentile criterion, e.g., 30%, participants with equal or greater than 30% performance were labeled as good performers, and those with less than 30% labeled as poor performers. The ABR measure, either wave V latency or amplitude, was then used to predict the good performers from the poor ones. The optimal value for each ABR parameter was determined by maximizing Youden's J statistic, with an AUC value of 0.5 corresponding to chance performance and 1.0 to perfect performance[70]. Both the raw and age-adjusted cognitive percentile scores were used as the predictive target.

The generalizability of the ABR measures and cognitive performance was also subject to linear discriminant analysis (LDA) using k-fold cross-validation[71]. The dataset was divided into k approximately equal-sized subsamples or folds, with k being 2, 5 or 10 in the present study. In each fold, k-1 subsamples served as the training data while the remaining subsample as the validation data. For example, using 5-fold cross validation with 118 participants, each fold contained 23–24 participants, and each training data contained 94–95 participants. For each cognitive performance, the LDA function was calculated for each fold, and the average over all folds was reported as the cross-validated estimate. At the extremes of cognitive performance like 1% or 99%, some folds contained only good or poor performers, leading to undefined sensitivity or specificity metrics. These folds were excluded from the average. The cross validation ensured that each observation was utilized for both training and validating the classifier, increasing the overall generalizability and reliability.

## Data availability

The source data for Figs. 1A, 2, 3, 4 and 5 can be found in Supplementary Data (in Excel format). Supplementary Materials include Supplementary Table 1 (The Association between cognition and ABR measures), Supplementary Table 2 (The Association of PTA and ABR measures), Supplementary Table 3 (The Association between the ABR-V latency and Cognitive Domains), Supplementary Table 4 (The Association between the ABR-V Amplitude and Cognitive Domains), Supplementary Table 5 (The Association between the hearing level or PTA and Cognitive Domains), Supplementary Fig. 1 (The area-under-curve (AUC) using ABR wave V parameters to predict age-unadjusted and adjusted cognitive performance), Supplementary Fig. 2 (Area-under-curve (AUC) analysis using 5-fold cross validation), and Supplementary Fig. 3 (Area-under-curve (AUC) analysis using 10-fold cross validation). All other data are availablee from the corresponding authors on reasonable request.

## Code availability

The ABR peak auto-detection program can be obtained by contacting the corresponding authors.

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

## Acknowledgements
The authors thank the UC Irvine Institute for Memory Impairments and Neurological Disorders (UCI MIND) for the assistance in the recruitment of participants, selection of cognitive tests and funding. The authors also thank three anonymous reviewers who not only provided helpful comments on the manuscript but also suggested additional sensitivity analysis and cross validation, which have strengthened the conclusions of the present study. This work was supported in part by NIH-NIDCD R01 DC015587, UCI MIND Innovative Research Project Award and McKnight Brain Research Foundation.

## Author contributions
Conceptualization and design: Y.H., C.B. and F.G.Z. Data collection: Y.H., Y.Y., J.V. and A.A. Data analysis: Y.H., Y.Y., M.M. and F.G.Z. Initial draft: Y.H. and F.G.Z. Editing and final approval: all authors.

## Competing interests
F.G.Z. owns stock in Axonics, Neocortix, Nurotron, Syntiant and Velox Biosystems. The other authors declare no competing interests.
