## [Transparent Peer Review file · Communications Biology]

Auditory Brainstem Responses as a Biomarker for Cognition

Corresponding Author: Professor Fan-Gang Zeng

Version 0:

Reviewer comments:

Reviewer #1

(Remarks to the Author)

This manuscript presents a large study investigating the association between ABR characteristics and cognition in a cross-sectional cohort. This work aims to improve our understanding of the reported relationship between hearing loss and cognitive impairment, which is poorly understood at the moment. The cohort provides a good foundation for investigating associations between auditory characteristics and cognition. The manuscript would benefit from additional sensitivity analyses e.g., investigating these associations within the elderly group, reducing the variance due to age, and reporting on the associations between the stimulus intensity and ABR characteristics. The statistical analyses are appropriate, but the researchers can consider omitting the univariate models from the results section. The discussion section is sometimes speculative and can be strengthened by focussing on the results of the current study. Overall a great study that can help shed light on the associations between characteristics of the auditory system and cognition.

Abstract

Line 2-3: Consider replacing “lacking” with “sparse” to not antagonize the Alzheimer’s disease research community. Or consider removing the ‘is lacking’ part of the sentence. There are instruments that are sensitive to early accumulation of Alzheimer’s Disease pathology related cognitive decline, such as the PACC (preclinical Alzheimer’s cognitive composite) questionnaire.

Line 8: Consider removing the phrase “not suprisingly” to increase readability of the sentence.

Introduction

Line 32: there are FDA approved treatments for AD pathology that work in terms of pathology clearance. It may be better to change the word “AD” to “AD-related dementia”, to indicate that there currently is no cure for dementia, the cognitive aspect of (among other neurodegenerative diseases) AD.

Line 37: more recently developed, and much more widely available, are plasma-based biomarkers of beta-amyloid and tau. It may be good to include these markers in addition to the mention of CSF-based markers, which are much rarer in clinical and research settings.

Line 39-41: Whereas PET and structural MRI have been successfully used as clinically relevant biomarkers of AD, functional MRI has only been applied in research settings and may not be considered a true biomarker as it generally cannot detect these patterns on an individual level. Perhaps consider restricting this sentence to methods that have given tangible biomarkers of AD (e.g., FDG-PET hypometabolism, amyloid PET, and tau PET).

Line 46: activities should probably be activity

Line 59: The causality of this relationship is not experimentally proven, and therefore it may be better to alter the phrasing throughout the introduction to prevent overstating the association between hearing loss and cognitive decline. The work in this manuscript is not dependent on a potential causal relationship between hearing loss and cognition.

Line 64 and 65: this statement is not supported by the literature. There have been numerous studies, including a large-scale randomized controlled trial, showing that exercise alters the trajectory of cognitive decline, even in individuals who are in a nursing or care home setting. See for instance: <https://alz-journals.onlinelibrary.wiley.com/doi/abs/10.1002/alz.069700>

Line 73: Perhaps consider replacing “understood” with “described”.

Results

Line 96: It seems superfluous to report the univariate analyses in the results section, because the authors know that age and hearing loss are significantly associated, and correctly take measures to control for age in their investigations of hearing loss and vice versa to prevent confounding of these variables. The authors can consider removing the univariate analyses from the results section, referring to the supplementary tables.

The reported age effect may be due to the large age range of the study. Whereas one would not expect cognitive impairment in the young group, without hearing loss, one would expect both hearing loss and some cognitive changes in the older group. The result of the study set up is that these variables are tightly correlated and the results show that the age effect is larger than the effect of hearing loss on cognition. To circumvent this to a certain degree, it would be interesting if the authors could repeat their regression models within the older age group only. Are there significant associations between hearing levels/ABR measures and cognition, adjusting for age, in this older group? This takes away some of the variance explained by age and focuses on the individuals who are more likely to have (subtle) cognitive impairment, and allows you to investigate what the role of hearing loss is in this association. Can the authors report on the results if these analyses are run with hearing thresholds as a categorical value indicating the presence/absence of hearing loss or the degree of hearing loss? These sensitivity analyses would strengthen the results section considerably.

Line 137-142: Figure 4. The latency used to generate the ROC curve in panel B is outside of the range of detected latencies in any of the groups, as the data in table 1 shows. Whereas a latency of >6.42 ms may detect low cognitive performers, this value is not present in the current dataset. Can the authors comment on this finding and the relevance to this dataset? Do the authors have a dataset with more severe cognitive impairment in which they can verify this cut-off for ABR wave V latency?

Discussion

It is not entirely clear whether the reported associations are true biomarkers and go beyond correlational, or alternatively strengthen the hypothesis that auditory system changes pose a risk factor for cognitive impairment. Additional analyses may help strengthen the proposed use of ABR as a biomarker of dementia.

Line 164 - 165: It would be interesting to include a table detailing the analysis on the cognitive subtests with the PTA instead of the ABR to strengthen the notion of the authors that this effect is specific to the ABR wave V.

Line 191: behavioal should probably read behavioral

Line 192 - 195: It is not immediately clear why the authors found no association between hearing levels and wave V parameters. Can the authors elaborate on this?

Line 200-202: This is highly speculative and not supported by the findings of the current study. The current study measured cognition and ABR at one time point, and it is currently unsupported that the ABR can be used to track the progression of cognitive decline.

Line 204-206: It is unclear why it would be useful to monitor the ABR of late stage AD patients. It is more likely that these factors are coupled in the early stages of the disease, whereas in late stage AD it is unclear what the added benefit of this measure is.

Methods

One potential caveat influencing the results may be the different inter-individual intensity levels of the click stimuli, which were set at 60 dB above detection threshold, meaning that these were of considerable higher intensity for the older age group. This is of particular interest since it is reported that the amplitude of wave V is determined by the stimulus intensity (not loudness) which drives synchronized neural activity (Rouillon et al., 2016). Have the authors considered sensitivity analyses comparing the intensity of the ABR stimuli to the amplitude of the ABR wave V?

Line 257: Can the authors describe the modality of the presentation of this test, was this done as an oral or a written presentation?

Line 259 has an extra comma.

Line 282: Can the authors describe how they normalized the cognitive scores?

Lines 292-293: Do the authors mean that they entered age as a covariate in the linear regression model?

Lines 295: Is the significance criterion of $p < 0.05$ adjusted for multiple comparisons? If not, this may inflate the type I error rate for the individual cognitive test scores in particular.

Line 296: Perhaps consider rephrasing the sentence to reflect that linear regression models are not able to answer questions on whether one variable predicts another. Regression models give information on whether there is an association, but the direction of the effect cannot be inferred. For that, longitudinal data and linear mixed effects models can provide more

insight. E.g.: To characterize the association between ABR characteristics and cognitive performance, etc.

Table 1:

Mean of the composite cognitive z-score is lower in the young group compared to the elderly group, is that correct? It looks as if the mean is not within the range given. The composite cognitive z-score is also provided twice in this table. It appears as if the middle-aged group performs better on some of the metrics than the young group (e.g., HINT-Quiet sentence). Have you looked at the distributions of the scores to see if this effect is driven by a few outliers, or whether there is another explanation for this?

Reviewer #2

(Remarks to the Author)

Study authors tested the hypothesis that the auditory brainstem response (ABR) predicts cognitive performance in adults. They enrolled 118 participants ranging in age from 18-92 years who completed a comprehensive battery of cognitive tests, audiometry, and ABR testing. They found that, even when controlling for age, superior cognitive performance was correlated with faster and larger ABR Wave V amplitudes, a measure of neural synchrony in the brainstem. They also found that ABR Wave V amplitude and latency was highly predictive of “good” and “poor” cognitive performance.

This is an innovative and provocative finding nicely presented in a concise form. Effect sizes are impressive and the paper will be of strong interest to diverse communities interested in aging, cognition, neurology, and hearing. It makes a nice advance forward in the literature on hearing and cognition, which frankly has produced many interesting but subtle findings.

My major concern is that the authors do not provide out-of-sample validation. The paper would be stronger if they fit their ROC analysis on a training subset of their data and validated it with AUCs on a test subset, or used a cross-validation method. Without this important addition there are concerns about overfitting and a lack of generalizability of their finding. (They could do something similar for the regression analyses, although this is less essential in my view.)

Additionally, I'm confused about the percentiles of cognitive performance in the ROC analysis. If these are straight transforms of the z-scores then they're highly correlated with age (see Figure 2A) in which case the authors are essentially reporting a correlation between ABRs and age. This would be more compelling if percentiles were age-corrected and referenced to norms.

Minor comments:

- Did any participants perform on the cognitive battery in a way to indicate mild cognitive impairment? If so, this could create an interesting secondary analysis.
- Check Table 1. Composite cognitive z-scores are presented twice, and it appears that young adults have a lower z-score than middle-aged adults. This seems inconsistent with the figures.
- The paper would benefit from one or two more rounds of careful copyediting for some typos and awkward sentences throughout.

Reviewer #3

(Remarks to the Author)

This study cross-sectionally examines links between ABR measures and cognition over a large age span in humans. They find that the cognition/hearing level link is mediated by age and that there are distinct links between ABR measures and cognition. While interesting, there are some methodological pieces that temper these findings and the contextualization of the results within the introduction and discussion are overreaching (e.g., this study provides no evidence that the ABR is a biomarker of AD). First, there is a large imbalance in the dataset with the greatest number of participants falling in the 'elderly' group and the largest variability occurring within this group. Thus, it is possible that the effects are being driven by the 'elderly' group. Secondly, the results showing that cognition and hearing level are explained by age seem to contradict the previous study discussed in the introduction showing that hearing aids can benefit those at risk of cognitive decline and other studies showing that hearing loss is a risk factor for cognitive decline – not to mention other previous studies that have used the ABR to estimate hearing threshold in individuals who cannot provide a behavioral response. Given the extant literature, it seems dubious that any link between ABR and cognition is not influenced at least to some extent by hearing level and that any link between hearing level and cognition is explained fully by age.

Major

Introduction/Discussion

This paper does not directly address Alzheimer's disease or biomarkers for the disease so it should be removed from the opening of the manuscript. Furthermore, the analyses, as currently written do not support some of the statements made in the discussion, such as the ABR being an early marker of cognitive decline (see my comments below).

Results/Methods

56% of the participants are in the 'elderly' group, while the remaining groups each make up 22% of the dataset. Is it possible to show that the cognitive/ABR relationships are not driven by the elderly group? It is important to demonstrate that this effect exists in each group independently to support the claim in the abstract and discussion that the ABR could provide an early indicator of cognitive decline.

The supplemental shows a relationship between PTA and Wave V Amplitude but this relationship is said to be irrelevant as it is suggested that it is explained by age. I struggle to reconcile this with what has been shown linking hearing aid use and cognition. As stated in the introduction, 'hearing aid use had a significant positive impact on individuals that were cognitively

healthy, but at increased risk for cognitive decline, suggesting that this intervention may decrease dementia risk in susceptible older individuals', it would be surprising if there really is no link between cognition and hearing level in these analyses, given that hearing aids work to counter the effects of hearing level. If age did indeed mediate the hearing level/cognition relationship, then everyone should benefit from wearing hearing aids as they get older. Given the large imbalance of older versus middle/younger adults in this dataset and the larger range of variability within the 'elderly' age group, it is possible that a relationship between ABR and hearing level that could not be accounted for by age would be seen with better balance across your groups. Otherwise, how do you reconcile this?

Provide a table with the average performance on the individual cognitive tests. Did any of the participants display any cognitive decline? If so, how old were they? Were any measures of cognitive function, such as the MOCA, used to set any inclusion or exclusion criterion based on cognitive abilities? Again, it is important to demonstrate that the relationship exists in people deemed 'not cognitively impaired' for this to be a marker of early detection, as described in the intro and discussion.

Why break the data into groups and what justification do you have for the group cutoffs? It is unclear from the text what analyses were done on the groups versus treating age as a continuous variable.

Minor

In the abstract, it is unclear whether 'hearing status' refers to the audiometric thresholds, the ABR, or both (thresholds were defined as 'hearing level', not 'hearing status' in figure 1).

In the abstract, without contextualization, it is unclear what is meant by the statement 'especially in low middle income countries'.

Line 73 'transit' should be 'transient'

Lines 90-93 mention results for speech in quiet and noise, but I don't see a description of those tests in the methods.

Version 1:

Reviewer comments:

Reviewer #1

(Remarks to the Author)

The reviewers addressed most of my concerns adequately, and the additions to the main body of the manuscript and supplementary materials are incredibly helpful for interpreting the results presented. However, I would like to stress once more the importance of making sure that readers understand that the univariate results are only included to represent the raw data, and the statistical outcomes of these analyses should not be interpreted since they are not corrected for confounders. If the authors prefer to keep the univariate results in the main body of the manuscript, it would be good to explicitly state in the manuscript that these results should not be interpreted as they are uncorrected for covariates which are significantly associated with the outcome of interest. It is, however, highly unusual to report on main effects without taking significantly associated covariates into account.

Reviewer #2

(Remarks to the Author)

Study authors have undertaken a thorough and thoughtful revision of the manuscript. The additional analyses are compelling and strengthen their conclusions. I also appreciate that they have softened some of the stronger statements in the initial submission and clarified that, while there is a conceptual link to AD, they are not testing for it per se. The paper reads well with compelling visuals and appropriate balance of the primary paper and supplement. Authors efforts are appreciated.

This paper should have an immediate and significant influence on the fields of neurology, aging, and hearing, spanning both basic and applied sciences. It is, in my opinion, highly appropriate for publication in its current form.

Reviewer #3

(Remarks to the Author)

The authors have done a nice job revising the manuscript based on my and the other reviewers' comments. I still have reservations about the manuscript that I believe should be addressed. These, as well as some minor comments and edits, are detailed below:

1. The focus on Alzheimer's Disease related dementia is unwarranted. Much of the introduction and discussion currently is devoted to the need for an early, objective measure of AD-related dementia, yet this paper does not include anyone with dementia, including AD-dementia, does not test anyone with a cognitive impairment, and does not follow the participants, who all have normal cognitive function, to see whether they develop cognitive impairment, dementia, or AD-dementia to determine whether the wave V latency or amplitude could predict eventual impairment. While testing the efficacy of wave V as an early biomarker of AD-dementia is a great future goal, and can be mentioned as a future goal in the section, the data at hand do not answer this question and it is misleading to the reader to put so much of the focus on that in the introduction and discussion.

2. Additionally, this study involved no source localization and so mention of the sources involved in this relationship between ABR and cognition should be removed from the discussion (the section beginning on line 257). The presumed generators of wave V are the lateral lemniscus and inferior colliculus, not the temporal lobe, which is where the author's ascribe the language and memory-related cognitive functions that are predicted by wave V latency and amplitude. It is not

clear why these presumed generators of the ABR and cognition would relate with one another and the current study cannot answer this.

3. There is no theoretical discussion of why V latency would be a better predictor of poor cognitive function, but V amplitude would better detect high.

4. Line 118 states 'We hypothesized that the ABR wave V, not wave I, would reflect subcortical neurological changes that are associated with corresponding cortical and cognitive changes'. Please elaborate on why wave V and not wave I.

5. Why was speech-in-quiet and speech-in-noise testing included if it was not considered in the analyses? What does it contribute to the story about ABRs and cognition? These measures need to be contextualized better.

6. While wave V may not align with hearing level, wave I has been shown to do so. To help anchor your data with the previous literature, does wave I show those same relations in your dataset?

7. Although age was accounted for, biological sex was not. There are known differences in auditory processing, as determined by the click ABR (see work by Jerger, for example). How do these influence the link between ABRs and cognition?

8. The methods say that the ABR was done on the 'better ear'. What was the degree of asymmetry of hearing loss between the two ears? How many had asymmetric hearing loss? An asymmetry between the two ears may be indicative of other forms of hearing loss besides presbycusis.

9. Why was the ABR collected on a single ear? The montage was setup to facilitate testing of both ears and would have provided a nice dataset for replication of the effect.

10. The methods say that the I-V IPL and V/I amplitude ratio were included 'as a relative measure to minimize individual differences in sex, age, and head geometry'. However, it was only wave V latency and amplitude that were found to be predictors of cognition (as stated on lines 167-170, ...the two combined measures (I-V latency difference and V/I amplitude ratio) produced inconsistent associations), they were not used as a potential biomarker for cognition). Age was found to partially explain some of the relationship between wave V and cognition. How can the authors be sure that sex and/or head geometry do not account for the remainder?

11. In the methods, split the cognitive tests so that each has their own section rather than a single long paragraph. It will be easier for the reader to see how each cognitive measure was assessed and what dependent variables were extracted from each test.

12. Line 429 says that the V/I amplitude was log-transformed to conform to a normal distribution. Were all the remaining DVs normally distributed or corrected to be normally distributed?

13. Line 308: remove the word 'special'

14. Provide the raw means and standard deviations for the cognition tests in the manuscripts. The mean and SD for the full sample is unnecessary in Table 2, because, by definition of the z-score, the mean is 0 and the SD 1.

15. The sentence that goes from line 461-463 needs to be corrected

16. Line 2: add an 'a' between 'suggested' and 'strong'

17. Line 9: 'age adjustment' should be 'adjusting for age'

18. Abstract: it's fine to keep the original last sentence, just remove 'especially in low middle income countries'

19. Line 157: 'Noted' should be 'Note'

20. Line 377: add an 'a' between 'with' and 'further'

Version 2:

Reviewer comments:

Reviewer #3

(Remarks to the Author)

The authors have done a very nice job responding to my comments.

A quick point of clarity, with respect to my comment in the last revision: "4. Line 118 states 'We hypothesized that the ABR wave V, not wave I, would reflect subcortical neurological changes that are associated with corresponding cortical and cognitive changes'. Please elaborate on why wave V and not wave I.":

I had meant for the authors to elaborate on why V and not I in the manuscript to give the reader some context when this hypothesis is mentioned. I recommend putting the first sentence of their response in the rebuttal letter: "Because wave I reflects the auditory nerve activity, whereas wave V reflects the brainstem and midbrain activity [and is] consistent with the animal work by Gray et al." into the manuscript where this hypothesis is mentioned.

Lastly, although I think there was a missed opportunity to collect responses to each ear as a way to replicate your analyses, given that all but one participant had symmetric hearing thresholds, I'm fine with only having results from a single ear -- as long as the one participant with asymmetric thresholds is not driving the results.

Reviewers' comments:

Reviewer #1 (Remarks to the Author):

This manuscript presents a large study investigating the association between ABR characteristics and cognition in a cross-sectional cohort. This work aims to improve our understanding of the reported relationship between hearing loss and cognitive impairment, which is poorly understood at the moment. The cohort provides a good foundation for investigating associations between auditory characteristics and cognition. The manuscript would benefit from additional sensitivity analyses e.g., investigating these associations within the elderly group, reducing the variance due to age, and reporting on the associations between the stimulus intensity and ABR characteristics. The statistical analyses are appropriate, but the researchers can consider omitting the univariate models from the results section. The discussion section is sometimes speculative and can be strengthened by focussing on the results of the current study. Overall a great study that can help shed light on the associations between characteristics of the auditory system and cognition.

Responses: Thank you for your helpful and constructive comments. We have addressed your concerns in our responses below.

Abstract

Line 2-3: Consider replacing “lacking” with “sparse” to not antagonize the Alzheimer’s disease research community. Or consider removing the ‘is lacking’ part of the sentence. There are instruments that are sensitive to early accumulation of Alzheimer’s Disease pathology related cognitive decline, such as the PACC (preclinical Alzheimer’s cognitive composite) questionnaire.

Responses: Replaced as suggested and the opening sentence has been reorganized in the revised abstract.

Line 8: Consider removing the phrase “not suprisingly” to increase readability of the sentence.

Responses: Removed.

Introduction

Line 32: there are FDA approved treatments for AD pathology that work in terms of pathology clearance. It may be better to change the word “AD” to “AD-related dementia”, to indicate that there currently is no cure for dementia, the cognitive aspect of (among other neurodegenerative diseases) AD.

Responses: Added.

Line 37: more recently developed, and much more widely available, are plasma-based biomarkers of beta-amyloid and tau. It may be good to include these markers in

addition to the mention of CSF-based markers, which are much rarer in clinical and research settings.

Line 39-41: Whereas PET and structural MRI have been successfully used as clinically relevant biomarkers of AD, functional MRI has only been applied in research settings and may not be considered a true biomarker as it generally cannot detect these patterns on an individual level. Perhaps consider restricting this sentence to methods that have given tangible biomarkers of AD (e.g., FDG-PET hypometabolism, amyloid PET, and tau PET.

Responses: We added a blood-based biomarker reference and removed fMRI while adding amyloid and tau PET description.

Line 46: activities should probably be activity

Responses: Changed.

Line 59: The causality of this relationship is not experimentally proven, and therefore it may be better to alter the phrasing throughout the introduction to prevent overstating the association between hearing loss and cognitive decline. The work in this manuscript is not dependent on a potential causal relationship between hearing loss and cognition.

Responses: Removed "causality" and greatly simplified this sentence in the revision.

Line 64 and 65: this statement is not supported by the literature. There have been numerous studies, including a large-scale randomized controlled trial, showing that exercise alters the trajectory of cognitive decline, even in individuals who are in a nursing or care home setting. See for instance: <https://alz-journals.onlinelibrary.wiley.com/doi/abs/10.1002/alz.069700>

Responses: Added this reference and softened the sentence in the revision.

Line 73: Perhaps consider replacing "understood" with "described".

Responses: Replaced.

Results

Line 96: It seems superfluous to report the univariate analyses in the results section, because the authors know that age and hearing loss are significantly associated, and correctly take measures to control for age in their investigations of hearing loss and vice versa to prevent confounding of these variables. The authors can consider removing the univariate analyses from the results section, referring to the supplementary tables.

Responses: We agree it seems superfluous to report the univariate analyses. We choose to preserve them in the revision because they not only represent the raw data but also set the stage for later analyses, including the sensitivity analysis suggested below.

The reported age effect may be due to the large age range of the study. Whereas one would not expect cognitive impairment in the young group, without hearing loss, one

would expect both hearing loss and some cognitive changes in the older group. The result of the study set up is that these variables are tightly correlated and the results show that the age effect is larger than the effect of hearing loss on cognition. To circumvent this to a certain degree, it would be interesting if the authors could repeat their regression models within the older age group only. Are there significant associations between hearing levels/ABR measures and cognition, adjusting for age, in this older group? This takes away some of the variance explained by age and focuses on the individuals who are more likely to have (subtle) cognitive impairment, and allows you to investigate what the role of hearing loss is in this association. Can the authors report on the results if these analyses are run with hearing thresholds as a categorical value indicating the presence/absence of hearing loss or the degree of hearing loss? These sensitivity analyses would strengthen the results section considerably.

Responses: We performed the sensitivity analysis as suggested and found that indeed, the observed association was primarily driven by the elderly group. We have added a paragraph in the Results section and highlighted this finding in the abstract and Discussion of the revised manuscript.

Line 137-142: Figure 4. The latency used to generate the ROC curve in panel B is outside of the range of detected latencies in any of the groups, as the data in table 1 shows. Whereas a latency of >6.42 ms may detect low cognitive performers, this value is not present in the current dataset. Can the authors comment on this finding and the relevance to this dataset? Do the authors have a dataset with more severe cognitive impairment in which they can verify this cut-off for ABR wave V latency?

Responses: There is likely misunderstanding here because Table 1 (row: ABR Wave V) shows that the range of V latency is 5.31-6.87 in both the full sample (2nd column) and the elderly (rightmost column). In other words, ">6.42 ms" is present in the present dataset. At present, we do not have any dataset with more severe cognitive impairment.

Discussion

It is not entirely clear whether the reported associations are true biomarkers and go beyond correlational, or alternatively strengthen the hypothesis that auditory system changes pose a risk factor for cognitive impairment. Additional analyses may help strengthen the proposed use of ABR as a biomarker of dementia.

Responses: We have conducted several additional analyses as suggested and they all point to the same direction. We have added an entire subsection in the Results section and highlighted this finding in the abstract and Discussion of the revised manuscript. The present result is still far from establishing the ABR as a true biomarker. Future longitudinal studies are needed, which has been acknowledged now in the discussion.

Line 164 - 165: It would be interesting to include a table detailing the analysis on the cognitive subtests with the PTA instead of the ABR to strengthen the notion of the authors that this effect is specific to the ABR wave V.

Responses: We included the suggested table as Supplementary Table 5, which showed no significant association between cognition and PTA once adjusted for age.

Line 191: behavioral should probably read behavioral

Responses: **Corrected.**

Line 192 - 195: It is not immediately clear why the authors found no association between hearing levels and wave V parameters. Can the authors elaborate on this?

Responses: **This lack of association is confirmed by your suggested analysis on association between cognition and PTA (Line 164-165 above). It is possible that the reduced wave V amplitude is not due to PTA but some other causes that are not reflected by thresholds, e.g., age-related cochlear synaptopathy (see also response to Reviewer 3). A reference is added in the revised Discussion.**

Line 200-202: This is highly speculative and not supported by the findings of the current study. The current study measured cognition and ABR at one time point, and it is currently unsupported that the ABR can be used to track the progression of cognitive decline.

Responses: **This sentence has been deleted.**

Line 204-206: It is unclear why it would be useful to monitor the ABR of late stage AD patients. It is more likely that these factors are coupled in the early stages of the disease, whereas in late stage AD it is unclear what the added benefit of this measure is.

Responses: **This sentence has been changed to "it is especially useful for monitoring special patients whose behavioral tasks are difficult or impossible to measure."**

Methods

One potential caveat influencing the results may be the different inter-individual intensity levels of the click stimuli, which were set at 60 dB above detection threshold, meaning that these were of considerable higher intensity for the older age group. This is of particular interest since it is reported that the amplitude of wave V is determined by the stimulus intensity (not loudness) which drives synchronized neural activity (Rouillon et al., 2016). Have the authors considered sensitivity analyses comparing the intensity of the ABR stimuli to the amplitude of the ABR wave V?

Responses: **We have not done this stimulus level sensitivity analysis. We have added the Rouillon et al. 2016 reference and the following sentence in the revision: "This relatively high click level was at or above the 60-dB knee point where the ABR wave V parameters became saturated with further increase in stimulus level ⁵⁷."**

Line 257: Can the authors describe the modality of the presentation of this test, was this done as an oral or a written presentation?

Responses: **Written.**

Line 259 has an extra comma.

Responses: **Removed.**

Line 282: Can the authors describe how they normalized the cognitive scores?

Responses: added “z-score” before normalized in the revision.

Lines 292-293: Do the authors mean that they entered age as a covariate in the linear regression model?

Responses: Age was adjusted only in multivariate analysis and this point was made clear in the revision.

Lines 295: Is the significance criterion of $p < 0.05$ adjusted for multiple comparisons? If not, this may inflate the type I error rate for the individual cognitive test scores in particular.

Responses: Yes, the significance criterion of $p < 0.05$ was adjusted for multiple comparisons (=by multiplying the unadjusted p by m , with m being the number of multiple comparisons).

Line 296: Perhaps consider rephrasing the sentence to reflect that linear regression models are not able to answer questions on whether one variable predicts another. Regression models give information on whether there is an association, but the direction of the effect cannot be inferred. For that, longitudinal data and linear mixed effects models can provide more insight. E.g.: To characterize the association between ABR characteristics and cognitive performance, etc.

Responses: Changed.

Table 1:

Mean of the composite cognitive z-score is lower in the young group compared to the elderly group, is that correct? It looks as if the mean is not within the range given. The composite cognitive z-score is also provided twice in this table.

It appears as if the middle-aged group performs better on some of the metrics than the young group (e.g., HINT-Quiet sentence). Have you looked at the distributions of the scores to see if this effect is driven by a few outliers, or whether there is another explanation for this?

Responses: Thanks for catching this typo: The mean in the young group should be 0.60 not -0.60. The extra cognitive score row is removed in the revised Table 1.

All HINT-Quiet scores have reached ceiling (=100% correct). Yes, the small difference between the young and mid-aged groups was due to one young subject performing 70%. Nevertheless, this outlier did not result in any significant group difference ($p = 0.17$).

Reviewer #2 (Remarks to the Author):

Study authors tested the hypothesis that the auditory brainstem response (ABR) predicts cognitive performance in adults. They enrolled 118 participants ranging in age from 18-92 years who completed a comprehensive battery of cognitive tests, audiometry, and ABR testing. They found that, even when controlling for age, superior cognitive performance was correlated with faster and larger ABR Wave V amplitudes, a measure of neural synchrony in the brainstem. They also found that ABR Wave V amplitude and latency was highly predictive of “good” and “poor” cognitive performance.

This is an innovative and provocative finding nicely presented in a concise form. Effect sizes are impressive and the paper will be of strong interest to diverse communities interested in aging, cognition, neurology, and hearing. It makes a nice advance forward in the literature on hearing and cognition, which frankly has produced many interesting but subtle findings.

My major concern is that the authors do not provide out-of-sample validation. The paper would be stronger if they fit their ROC analysis on a training subset of their data and validated it with AUCs on a test subset, or used a cross-validation method. Without this important addition there are concerns about overfitting and a lack of generalizability of their finding. (They could do something similar for the regression analyses, although this is less essential in my view.)

Responses: Thank you for this constructive comment. We have performed cross-validation as suggested and added a new subsection in the results (Figure 5 and Supplementary Fig. 2 and 3) and methods sections. The cross-validation produced a similar pattern of result, hopefully alleviating your concerns about overfitting and a lack of generalizability.

Additionally, I’m confused about the percentiles of cognitive performance in the ROC analysis. If these are straight transforms of the z-scores then they’re highly correlated with age (see Figure 2A) in which case the authors are essentially reporting a correlation between ABRs and age. This would be more compelling if percentiles were age-corrected and referenced to norms.

Responses: We performed age-adjusted analysis as suggested and indeed, the result (Fig. 4B and 5B) still showed a significant, but slightly reduced correlation between wave V parameters and cognition. This consistent pattern of results suggests that the ABR wave V is correlated with two components of cognition, with one being age-dependent and other age-independent. This is an important and compelling result. We have highlighted it in the revision.

Minor comments:

- Did any participants perform on the cognitive battery in a way to indicate mild cognitive impairment? If so, this could create an interesting secondary analysis.

Responses: No, we did not perform any screening test. However, participants were asked if they had been exposed to the cognitive tests previously including if they had a formal diagnosis of MCI or Dementia. We added the following statement to the Methods- participants section “The elderly participants were recruited without clinically diagnosed mild cognitive decline or ADRD.”

- Check Table 1. Composite cognitive z-scores are presented twice, and it appears that young adults have a lower z-score than middle-aged adults. This seems inconsistent with the figures.

Responses: Thank you for catching the repetition and the typo, both have been corrected in the revision.

- The paper would benefit from one or two more rounds of careful copyediting for some typos and awkward sentences throughout.

Responses: We have carefully proofread the revision as suggested. All changes are marked in red text.

Reviewer #3 (Remarks to the Author):

This study cross-sectionally examines links between ABR measures and cognition over a large age span in humans. They find that the cognition/hearing level link is mediated by age and that there are distinct links between ABR measures and cognition. While interesting, there are some methodological pieces that temper these findings and the contextualization of the results within the introduction and discussion are overreaching (e.g., this study provides no evidence that the ABR is a biomarker of AD). First, there is a large imbalance in the dataset with the greatest number of participants falling in the 'elderly' group and the largest variability occurring within this group. Thus, it is possible that the effects are being driven by the 'elderly' group. Secondly, the results showing that cognition and hearing level are explained by age seem to contradict the previous study discussed in the introduction showing that hearing aids can benefit those at risk of cognitive decline and other studies showing that hearing loss is a risk factor for cognitive decline – not to mention other previous studies that have used the ABR to estimate hearing threshold in individuals who cannot provide a behavioral response. Given the extant literature, it seems dubious that any link between ABR and cognition is not influenced at least to some extent by hearing level and that any link between hearing level and cognition is explained fully by age.

Responses: Thank you for your helpful and constructive comments. We have addressed your concerns in the response below.

Major

Introduction/Discussion

This paper does not directly address Alzheimer's disease or biomarkers for the disease so it should be removed from the opening of the manuscript. Furthermore, the analyses, as currently written do not support some of the statements made in the discussion, such as the ABR being an early marker of cognitive decline (see my comments below).

Responses: We agree that this paper does not directly address AD. To address your concern, we have revised the beginning sentence in both Abstract and Introduction, deleted treatment-related sentences and references in Introduction, and explicitly acknowledged this limitation in the last paragraph of Discussion. We have preserved the remaining AD-related introduction to provide the context for the present and future studies.

Results/Methods

56% of the participants are in the 'elderly' group, while the remaining groups each make up 22% of the dataset. Is it possible to show that the cognitive/ABR relationships are not driven by the elderly group? It is important to demonstrate that this effect exists in each group independently to support the claim in the abstract and discussion that the ABR could provide an early indicator of cognitive decline.

Responses: We have performed sensitivity analysis based on the three groups (see also response to Reviewer 1 who had expressed a similar concern). We have added a Results subsection to describe the main finding that this cognitive/ABR association was mostly

driven by the elderly group, to a less extent by the middle-aged group, and not at all by the young group. This pattern of result does not show the association in each group but makes sense because cognitive decline is unlikely to occur in the young group.

The supplemental shows a relationship between PTA and Wave V Amplitude but this relationship is said to be irrelevant as it is suggested that it is explained by age. I struggle to reconcile this with what has been shown linking hearing aid use and cognition. As stated in the introduction, 'hearing aid use had a significant positive impact on individuals that were cognitively healthy, but at increased risk for cognitive decline, suggesting that this intervention may decrease dementia risk in susceptible older individuals', it would be surprising if there really is no link between cognition and hearing level in these analyses, given that hearing aids work to counter the effects of hearing level. If age did indeed mediate the hearing level/cognition relationship, then everyone should benefit from wearing hearing aids as they get older. Given the large imbalance of older versus middle/younger adults in this dataset and the larger range of variability within the 'elderly' age group, it is possible that a relationship between ABR and hearing level that could not be accounted for by age would be seen with better balance across your groups. Otherwise, how do you reconcile this?

Responses: We tried to reconcile this in the following way. First, we used an individualized relatively high stimulation level (60 dB above each individual threshold), which minimized the effect of hearing level or loss if any. Second, it is possible that the reduced wave V amplitude is not due to PTA but some other causes like cochlear synaptopathy. We have emphasized the use of this high-level ABR in the abstract and ABR methods (see Rouillon et al. reference) and added Supplementary Table 5 and a synaptopathy reference in Discussion of the revised manuscript (see also response to Reviewer 1).

Yes, it would be interesting for future studies to examine longitudinally whether hearing aid use would slow down the ABR deterioration, corresponding to its positive impact on cognitive decline. We have added a sentence to address this important future study in Discussion (end of the section on "ABR-cognition is specific to cognitive domains").

Provide a table with the average performance on the individual cognitive tests. Did any of the participants display any cognitive decline? If so, how old were they? Were any measures of cognitive function, such as the MOCA, used to set any inclusion or exclusion criterion based on cognitive abilities? Again, it is important to demonstrate that the relationship exists in people deemed 'not cognitively impaired' for this to be a marker of early detection, as described in the intro and discussion.

Responses: The individual tests are provided as Table 2 in the main text of the revised manuscript. Another version of Table 2 lists the raw scores and is attached below for your information:

Table 2. The participant 's cognitive specific domain scores

Cognitive Test mean (SD), range	Full-sample	Young n= 26	Middle-Aged n= 26	Elderly n= 66
Word Learning	22.10 (3.59), 12– 29	24.08 (2.54), 19– 29	22.92 (3.31), 15– 27	21.00 (3.66), 12– 28
Delayed recall	7.08 (2.01), 1– 10	8.42 (1.33), 6– 10	7.31 (1.89), 3– 10	6.47 (2.02), 1– 10
Word Recognition	19.52 (1.31), 10– 20	19.92 (0.39), 18– 20	19.81 (0.40), 19– 20	19.24 (1.67), 10– 20
Animal Fluency	22.81 (5.65), 9– 36	25.73 (5.58), 11– 34	23.31 (4.57), 14– 32	21.47 (5.66), 9– 36
TMT A (Executive), seconds	26.77 (8.84), 12-57	20.48 (4.70), 12-32	25.15 (7.29), 15-39	29.89 (9.22), 14-57
TMT B (Executive), seconds	66.65 (27.33), 30– 218	50.35 (13.72), 30– 84	61.62 (21.42), 36– 134	75.05 (30.13), 37– 218
4 MT accuracy (spatial)	10.10 (2.87), 2– 15	11.85 (2.11), 5– 15	10.50 (2.87), 2– 14	9.26 (2.82), 3– 14
4 MT RT (spatial), mseconds	8150.77 (2221.63), 3968– 13810	7421.54 (2033.72), 4007– 11627	7538.85 (1736.56), 3968– 10305	8679.11 (2342.56), 4474– 13810
Visual Discrimination	15.97 (1.82), 11– 18	17.27 (0.96), 14– 18	15.92 (2.04), 11– 18	15.48 (1.76), 11– 18
SDMT (Attention/ working memory)	52.42 (11.63), 21– 79	64.54 (9.24), 40– 79	52.73 (8.94), 35– 72	47.53 (9.84), 21– 66

To address your concerns above, we have also added the following sentence in Discussion (“ABR as a cognitive biomarker: Clinical utilities”):

“Although the present study did not include any clinically diagnosed patients with mild cognitive impairment or ADRD, we found two participants to have mild cognitive impairment based on previously-published normative cognitive data ⁵⁴. Both participants were in the elderly group (84 and 92 years) and had wave latency (6.47 and 6.87 ms, respectively) longer than the presently-proposed 6.42-ms criterion for identifying poor cognitive performers.”

Why break the data into groups and what justification do you have for the group cutoffs? It is unclear from the text what analyses were done on the groups versus treating age as a continuous variable.

Responses: There were two justifications for breaking the data into three groups. First, similar breakups were used in a previous study that collected population normative data (Hankee et al. 2016). Second, the three groups allowed visualization of interactions of the age factor with other factors (e.g., with hearing loss in Fig. 2B,D and with ABR parameters in Fig. 3). Without these age-related, color-coded symbols, it is impossible to see such interactions. Both justifications are now stated in the revision.

Yes, the age was still analyzed as a continuous variable. We have explicitly stated this fact in the revised Methods section.

Minor

In the abstract, it is unclear whether 'hearing status' refers to the audiometric thresholds, the ABR, or both (thresholds were defined as 'hearing level', not 'hearing status' in figure 1).

Responses: **changed to "hearing level"**.

In the abstract, without contextualization, it is unclear what is meant by the statement 'especially in low middle income countries'.

Responses: **Deleted**.

Line 73 'transit' should be 'transient'

Responses: **Corrected**.

Lines 90-93 mention results for speech in quiet and noise, but I don't see a description of those tests in the methods.

Responses: **A speech recognition method section was added in the revision**.

Reviewers' comments:

Reviewer #1 (Remarks to the Author):

The reviewers addressed most of my concerns adequately, and the additions to the main body of the manuscript and supplementary materials are incredibly helpful for interpreting the results presented. However, I would like to stress once more the importance of making sure that readers understand that the univariate results are only included to represent the raw data, and the statistical outcomes of these analyses should not be interpreted since they are not corrected for confounders. If the authors prefer to keep the univariate results in the main body of the manuscript, it would be good to explicitly state in the manuscript that these results should not be interpreted as they are uncorrected for covariates which are significantly associated with the outcome of interest. It is, however, highly unusual to report on main effects without taking significantly associated covariates into account.

Responses: Thank you again for your helpful and constructive comments. We have added the following sentence to explicitly state this fact: "Note that this significant association revealed by the univariate analysis is uncorrected for covariates and is included to represent the raw data."

Reviewer #2 (Remarks to the Author):

Study authors have undertaken a thorough and thoughtful revision of the manuscript. The additional analyses are compelling and strengthen their conclusions. I also appreciate that they have softened some of the stronger statements in the initial submission and clarified that, while there is a conceptual link to AD, they are not testing for it per se. The paper reads well with compelling visuals and appropriate balance of the primary paper and supplement. Authors efforts are appreciated.

This paper should have an immediate and significant influence on the fields of neurology, aging, and hearing, spanning both basic and applied sciences. It is, in my opinion, highly appropriate for publication in its current form.

Responses: Thank you for your helpful and supportive comments.

Reviewer #3 (Remarks to the Author):

The authors have done a nice job revising the manuscript based on my and the other reviewers' comments. I still have reservations about the manuscript that I believe should be addressed. These, as well as some minor comments and edits, are detailed below:

Responses: Thank you again for your helpful and constructive comments. We have addressed your concerns in the response below.

1. The focus on Alzheimer's Disease related dementia is unwarranted. Much of the introduction and discussion currently is devoted to the need for an early, objective measure of AD-related dementia, yet this paper does not include anyone with dementia, including AD-dementia, does not test anyone with a cognitive impairment, and does not follow the participants, who all have normal cognitive function, to see whether they develop cognitive impairment, dementia, or AD-dementia to determine whether the wave V latency or amplitude could predict eventual impairment. While testing the efficacy of wave V as an early biomarker of AD-dementia is a great future goal, and can be mentioned as a future goal in the section, the data at hand do not answer this question and it is misleading to the reader to put so much of the focus on that in the introduction and discussion.

Responses: We have re-written the introduction to focus on cognition rather than AD as suggested.

2. Additionally, this study involved no source localization and so mention of the sources involved in this relationship between ABR and cognition should be removed from the discussion (the section beginning on line 257). The presumed generators of wave V are the lateral lemniscus and inferior colliculus, not the temporal lobe, which is where the author's ascribe the language and memory-related cognitive functions that are predicted by wave V latency and amplitude. It is not clear why these presumed generators of the ABR and cognition would relate with one another and the current study cannot answer this.

Responses: We agree that the current study cannot answer why the presumed generators of the ABR and cognition are related to each other. We think this relationship is an important finding worth mentioning and future investigation. We have added a new paragraph in Discussion to address this issue and related comment #3: "However, it is not clear why the ABR wave V parameters can predict individual cognitive performance. On surface, the neural structures underlying the ABR wave V and language and memory-related cognitive functions are entirely different, with the former being the lateral lemniscus and inferior colliculus 34 and the latter being the temporal lobe and hippocampus 42,43. There are likely common neural mechanisms from the brainstem to cortex that are responsible for this ABR-cognition association. We

propose two possible such mechanisms based on the present result showing that the wave V latency is a better predictor of poor cognitive function, but the wave V amplitude predicts high. One mechanism is related to synaptic delay and demyelination that produce longer wave V latency and poorer cognitive performance, whereas the other mechanism is related to total neuronal health and synchrony that produce higher wave V amplitude and better cognitive performance 35. Future investigation is needed to clarify this ABR-cognition relationship and its underlying neural mechanisms..”

3. There is no theoretical discussion of why V latency would be a better predictor of poor cognitive function, but V amplitude would better detect high.

Responses: *see above.*

4. Line 118 states ‘We hypothesized that the ABR wave V, not wave I, would reflect subcortical neurological changes that are associated with corresponding cortical and cognitive changes’. Please elaborate on why wave V and not wave I.

Responses: *Because wave I reflects the auditory nerve activity, whereas wave V reflects the brainstem and midbrain activity as stated in the previous paragraph. This hypothesis is also consistent with the animal work by Gray et al. Both contexts were discussed in the introduction. We added “also” in the sentence to make a stronger connection to these contexts.*

5. Why was speech-in-quiet and speech-in-noise testing included if it was not considered in the analyses? What does it contribute to the story about ABRs and cognition? These measures need to be contextualized better.

Responses: *Speech testing results are included as part of standard hearing measures like pure-tone thresholds. We added the following sentence to contextualize the inclusion: “This age-dependent hearing threshold elevation is partially consistent with suprathreshold speech recognition measures.”*

6. While wave V may not align with hearing level, wave I has been shown to do so. To help anchor your data with the previous literature, does wave I show those same relations in your dataset?

Responses: *No, wave I was not associated with hearing level –please see supplementary table 2 first 2 rows. This is expected as we presented the stimuli at an equal sensation level (as opposed to previous studies at a fixed stimulus level). We added “or hearing level (Supplementary Table 2)” in the results section.*

7. Although age was accounted for, biological sex was not. There are known differences in auditory processing, as determined by the click ABR (see work by Jerger, for example). How do these influence the link between ABRs and cognition?

Responses: We found that sex is only associated with ABR V measures but not associated with cognitive performance. We acknowledge that future studies need to include sex and other variables: “Another limitation is that the present study did not consider other potentially important variables like sex, head size, education and socioeconomic status. A much larger sample size than the current one would be needed to delineate the contributions of these variables to the ABR-cognition relationship.”

8. The methods say that the ABR was done on the ‘better ear’. What was the degree of asymmetry of hearing loss between the two ears? How many had asymmetric hearing loss? An asymmetry between the two ears may be indicative of other forms of hearing loss besides presbycusis.

Responses: On a group level, the hearing loss is symmetrical as the mean difference in PTA between ears was 1 dB (SD= 16 dB). The large SD was driven by one participant who had single-sided deafness. We chose the better ear as done in previous studies addressing hearing and cognition (see below).

Lin, F. R. (2011). Hearing loss and cognition among older adults in the United States. *J. Gerontol. A Biol. Sci. Med. Sci.* 66, 1131–1136. doi: 10.1093/gerona/qlr115

Lin, F. R., Ferrucci, L., Metter, E. J., An, Y., Zonderman, A. B., and Resnick, S. M. (2011a). Hearing loss and cognition in the baltimore longitudinal study of aging. *Neuropsychology* 25:763. doi: 10.1037/a0024238

Lin, F. R., Metter, E. J., O’Brien, R. J., Resnick, S. M., Zonderman, A. B., and Ferrucci, L. (2011b). Hearing loss and incident dementia. *Arch. Neurol.* 68, 214–220.

Deal, J. A., Betz, J., Yaffe, K., Harris, T., Purchase-Helzner, E., Satterfield, S., et al. (2017). Hearing impairment and incident dementia and cognitive decline in older adults: the health ABC study. *J. Gerontol. Ser. A* 72, 703–709.

Deal, J. A., Sharrett, A. R., Albert, M. S., Coresh, J., Mosley, T. H., Knopman, D., et al. (2015). Hearing impairment and cognitive decline: a pilot study conducted within the atherosclerosis risk in communities neurocognitive study. *Am. J. Epidemiol.* 181, 680–690. doi: 10.1093/aje/kwu333

Hamza Y and Zeng FG (2021) Tinnitus Is Associated With Improved Cognitive Performance in Non-hispanic Elderly With Hearing Loss. *Front. Neurosci.* 15:735950. doi: 10.3389/fnins.2021.735950

9. Why was the ABR collected on a single ear? The montage was setup to facilitate testing of both ears and would have provided a nice dataset for replication of the effect.

Responses: We chose the better ear so that we could present the stimulus at the same sensation level (to minimize the peripheral hearing loss effect and other potential confounding factors like asymmetrical hearing and binaural interactions).

10. The methods say that the I-V IPL and V/I amplitude ratio were included ‘as a relative measure to minimize individual differences in sex, age, and head geometry’. However, it

was only wave V latency and amplitude that were found to be predictors of cognition (as stated on lines 167-170, ...the two combined measures (I-V latency difference and V/I amplitude ratio) produced inconsistent associations), they were not used as a potential biomarker for cognition). Age was found to partially explain some of the relationship between wave V and cognition. How can the authors be sure that sex and/or head geometry do not account for the remainder?

Responses: Both sex and head size can possible account for the individual variability. We acknowledge this limitation in Discussion (see response to Comment #7).

11. In the methods, split the cognitive tests so that each has their own section rather than a single long paragraph. It will be easier for the reader to see how each cognitive measure was assessed and what dependent variables were extracted from each test.

Responses: Done.

12. Line 429 says that the V/I amplitude was log-transformed to conform to a normal distribution. Were all the remaining DVs normally distributed or corrected to be normally distributed?

Responses: Yes.

13. Line 308: remove the word 'special'

14. Provide the raw means and standard deviations for the cognition tests in the manuscripts. The mean and SD for the full sample is unnecessary in Table 2, because, by definition of the z-score, the mean is 0 and the SD 1.

15. The sentence that goes from line 461-463 needs to be corrected

16. Line 2: add an 'a' between 'suggested' and 'strong'

17. Line 9: 'age adjustment' should be 'adjusting for age'

18. Abstract: it's fine to keep the original last sentence, just remove 'especially in low middle income countries'

19. Line 157: 'Noted' should be 'Note'

20. Line 377: add an 'a' between 'with' and 'further'

Responses: All suggestions and comments from 15 to 20 have been adopted. Thank you again for your careful and helpful review.